# Complement Component C1q as a Potential Diagnostic Tool for Myalgic Encephalomyelitis/Chronic Fatigue Syndrome Subtyping

**DOI:** 10.3390/jcm10184171

**Published:** 2021-09-15

**Authors:** Jesús Castro-Marrero, Mario Zacares, Eloy Almenar-Pérez, José Alegre-Martín, Elisa Oltra

**Affiliations:** 1ME/CFS Research Unit, Division of Rheumatology, Vall d’Hebron Research Institute, Universitat Autònoma de Barcelona, 08035 Barcelona, Spain; 2Departamento de Ciencias Básicas y Transversales, Facultad de Veterinaria y Ciencias Experimentales, Universidad Católica de Valencia San Vicente Mártir, 46001 Valencia, Spain; mario.zacares@ucv.es; 3Centro de Investigación Traslacional San Alberto Magno, Universidad Católica de Valencia San Vicente Mártir, 46001 Valencia, Spain; eloy.almenar@ucv.es; 4ME/CFS Clinical Unit, Division of Rheumatology, Vall d’Hebron University Hospital, Universitat Autònoma de Barcelona, 08035 Barcelona, Spain; jalegre@vhebron.net; 5Department of Pathology, School of Health Sciences, Universidad Católica de Valencia San Vicente Mártir, 46001 Valencia, Spain

**Keywords:** myalgic encephalomyelitis, chronic fatigue syndrome, C1q, complement system, blood analytics, diagnosis, symptoms, cluster analysis

## Abstract

Background: Routine blood analytics are systematically used in the clinic to diagnose disease or confirm individuals’ healthy status. For myalgic encephalomyelitis/chronic fatigue syndrome (ME/CFS), a disease relying exclusively on clinical symptoms for its diagnosis, blood analytics only serve to rule out underlying conditions leading to exerting fatigue. However, studies evaluating complete and large blood datasets by combinatorial approaches to evidence ME/CFS condition or detect/identify case subgroups are still scarce. Methods: This study used unbiased hierarchical cluster analysis of a large cohort of 250 carefully phenotyped female ME/CFS cases toward exploring this possibility. Results: The results show three symptom-based clusters, classified as severe, moderate, and mild, presenting significant differences (*p* < 0.05) in five blood parameters. Unexpectedly the study also revealed high levels of circulating complement factor C1q in 107/250 (43%) of the participants, placing C1q as a key molecule to identify an ME/CFS subtype/subgroup with more apparent pain symptoms. Conclusions: The results obtained have important implications for the research of ME/CFS etiology and, most likely, for the implementation of future diagnosis methods and treatments of ME/CFS in the clinic.

## 1. Introduction

Myalgic encephalomyelitis/chronic fatigue syndrome (ME/CFS) constitutes a serious health problem that truncates the life of millions of people and their families around the world [1,2,3]. ME/CFS is a chronic condition characterized by profound fatigue which is exacerbated by physical/mental and emotional activity (also known as PEM; post-exertional malaise), lack of refreshing sleep and dysautonomia, and multiple additional comorbidities [4]; its diagnosis still solely relies on clinical symptom assessment [5,6,7] after ruling out potential subjacent illness that could explain patient´s symptoms.

Despite a number of studies aimed at evidencing routine clinical parameters that may be useful, at least for the suspicion of an ME/CFS case, few are the differences that have been reported [8]. For example, Nacul et al. found significantly lower median values of serum creatine kinase (CK) in severely ill patients compared to healthy controls (HCs) and non-severe ME/CFS (median = 54, 101.5, and 84 U/L, respectively) [9], a finding confirmed by two additional studies [10,11]. While CK differences may be derived from patient sedentarism itself, some potential differences, including the levels of alkaline phosphatase, free T4 levels, or eosinophil counts, detected at lower significance (*p* < 0.1) in small cohorts (*n* = 15/group) [10] deserve further exploration in larger cohorts, individually or in combination with others.

Blood factors differentially altered in ME/CFS subgroups may constitute valuable tools in the clinic for achieving improved patient treatments, particularly for precision medicine purposes, while they may also serve to minimize patient heterogeneity in research studies. Unveiling the nature of ME/CFS, in fact, might well depend on homogeneous patient subset assessment, boosting the statistical robustness of data. 

Therefore, in the current study we aimed at identifying clinical parameters that differentiate ME/CFS case subgroups by themselves or in relation to symptom severity, in a large cohort of female ME/CFS cases (*n* = 250), with potential therapeutic and/or research purposes.

## 2. Materials and Methods

### 2.1. Participants

In this observational, single-center, cross-sectional cohort study, a total of 250 females with ME/CFS were consecutively referred to a tertiary care referral center for clinical evaluation by a ME/CFS specialized clinician (Vall d’Hebron University Hospital, Barcelona, Spain) between March 2017 and December 2019. Participants were invited after eligibility was confirmed. Inclusion criteria consisted in adult female individuals fulfilling the 1994 CDC/Fukuda definition [5] and 2003 Canadian Consensus Criteria for ME/CFS [6]. Participants were excluded if they were previously diagnosed with any serious illnesses or comorbid diseases that could be associated with their symptoms. Participants donated a blood sample for routine blood testing, filled out validated standardized questionnaires, and provided demographic data and clinical characteristics at the time of their inclusion in the study.

The study procedures were reviewed and approved in accordance with the recommendations from the local Clinical Research Ethics Committee (Vall d’Hebron University Hospital, Barcelona, Spain; IRB protocol number: CEIC/PR-AG-VITAE-2015, approved in June 2015). All subjects voluntarily provided written signed informed consent prior to study participation, according to the guidelines of the Declaration of Helsinki and in compliance with current Spanish regulations on clinical research and the standards of EU good clinical practice.

### 2.2. Measures

Participants were asked to fill out validated self-reported outcome measures as symptom assessment tools. The measures described below were used to evaluate all participants under the supervision of two trained investigators (J.C.-M. and J.A.), who oversaw participant compliance.

#### 2.2.1. Fatigue Impact Scale

The Fatigue Impact Scale (FIS-40) is a 40-item questionnaire designed to assess fatigue symptoms as part of an underlying chronic condition. It includes three domains reflecting the perceived feeling of fatigue: physical (10 items), cognitive (10 items), and psychosocial functions (20 items). Each item is scored from zero (no fatigue) to four (severe fatigue). The overall score is calculated by adding together the responses to the 40 questions (ranging from 0 to 160 points). Higher scores indicate more functional limitations due to fatigue [12].

#### 2.2.2. Composite Autonomic Symptom Score

For measuring autonomic dysfunction, all participants were screened using the Composite Autonomic Symptom Score (COMPASS-31), a 31-item refined and abbreviated questionnaire designed to evaluate the frequency and severity of autonomic function symptoms, grouped in six domains: orthostatic intolerance (four items), vasomotor (three items), secretomotor (four items), gastrointestinal (12 items), bladder (three items), and pupillomotor symptoms (five items). Added together, the six domain scores provide a total COMPASS-31 score ranging from 0 to 100 points. Higher scores indicate more severe autonomic complaints [13].

#### 2.2.3. Pittsburgh Sleep Quality Index

The Pittsburgh Sleep Quality Index (PSQI) is a 19-item self-administrated questionnaire commonly used to assess sleep disturbances over a 1 month interval. Scores are acquired on each of the seven domains of sleep quality: subjective sleep quality, sleep latency, sleep duration, habitual sleep efficiency, sleep disturbances, use of sleeping medication, and daytime dysfunction. Each domain is scored from zero to three (zero = no problems and three = severe problems). The overall PSQI score ranges from 0 to 21 points, with scores ≥ 5 indicating poorer sleep quality [14].

#### 2.2.4. Short-Form-36 Health Survey

The SF-36 Health Survey questionnaire, a generic scale that provides a health status profile, was used to assess quality of life. The SF-36 comprises 36 questions which explore eight dimensions of health status (physical function, role limitations due to physical health, bodily pain, general health, vitality, social functioning, emotional role, and mental health), as well as two general subscales covering the physical and mental health domains [15].

### 2.3. Blood Collection and Processing

Blood samples were collected via venipuncture after a 12 h overnight fasting for immediate routine lab tests by an experienced research nurse at the ME/CFS outpatient clinic (Vall d’Hebron University Hospital, Barcelona, Spain). Blood samples were delivered to the local core laboratory at the hospital within 2 h of collection and analyzed consecutively. Standard HUVH (Vall d’Hebron University Hospital Core Lab, Barcelona, Spain) laboratory protocols were used for the collection, transport and processing, and routine blood tests following standard operating procedures (SOPs).

### 2.4. Blood Analytics

Baseline laboratory tests were used primarily to exclude primary ME/CFS symptoms of other fatigue-related conditions. These fasting blood tests comprised full blood count, erythrocyte sedimentation rate (ESR), platelets, blood biochemistry parameters, creatinine, fasting glucose, urea, uric acid, bilirubin, electrolyte test (sodium, potassium, calcium), liver function tests (AST, ALT, ALP, GGT), lipid profile (cholesterol, triglycerides, LDL, HDL), thyroid function tests, vitamin D, immunoglobulin (IgA, IgM, IgG, and their isotypes), complement proteins (C1 inhibitor, C1q, C3, C4), and anti-phospholipid antibodies (cardiolipin, beta-2-glycoprotein I). Complement levels were measured by nephelometry using a BN II System (Siemens Healthcare Headquarters, Erlangen, Germany). Normal range reference levels provided by the Vall d’Hebron University Hospital, Barcelona, Spain for each studied variable are shown in Appendix A.

### 2.5. Cluster Analysis

Hierarchical cluster analysis was initially conducted using Ward’s method [16] to identify the number of clusters chosen on the basis of interpretability and usefulness. A k-means cluster analysis was then carried out to assign participants to clusters. Welch’s ANOVA was used on variables of interest to analyze differences among clusters, followed by pairwise comparisons between clusters when the univariate analyses were significant (*p* < 0.05).

### 2.6. Statistical Analysis and Plotting

Continuous data are shown as means ± SD (standard deviation). Statistical differences were determined using two-tailed unpaired Welch *t*-tests. Normal distribution was assessed by the Shapiro–Wilk normality test. Categorical variables are presented as *n* (%), normally distributed variables are presented as mean ± SD, and non-normally distributed variables are presented as median (interquartile range). Differences between groups were considered significant at *p* ≤ 0.05. Variable correlations were evaluated by the simple linear regression method (least-squares approach). Statistical analyses were conducted with R 3.6.3 [17], and figures were produced using the package ggplot2 [18].

## 3. Results

### 3.1. Demographics and Clinical Characteristics of the Participants

This prospective observational study included 250 adult females diagnosed with ME/CFS by 1994 CDC/Fukuda and 2003 CCC [5,6], the analysis of 69 laboratory blood tests, demographic variables, and four validated self-reported questionnaires to assess disease severity and comorbidities [12,13,14,15]. In addition, cardiac variables and medication prescriptions were also recorded (Appendix A). Table 1 shows descriptive parameters of the study participants. The average age of participants was 45.9 ± 7.02 years, 11.6% (29/250) presented obesity (BMI ≥ 30), fitting with the 13% assessed for the general population [19], average heart rate was 78.5 ± 10.3 bpm, and systolic and diastolic blood pressure were 125.8 ± 2.5 and 76.3 ± 1.6, respectively, among participants. The overall FIS-40 score range reflected the presence of participants with different degrees of fatigue severity. The vast majority of participants had a severe fatigue score (98.8%) while only 1.2% had mild/moderate fatigue, as assessed by the FIS-40 questionnaire provided to the study participants. In addition, over 50% of subjects were taking at least more than one medication as usual/routine care treatment (Table 1).

### 3.2. Exploratory Case Cluster Analysis Based on Symptoms

After applying unbiased hierarchical clustering and optimal grouping based on k-means screenings to identify case clusters, as detailed in Section 2, a set of three clusters showing significant differences in their total FIS-40, total COMPASS-31, total PSQI, physical functioning, and bodily pain scores was obtained (Table 2). Plotting of the itemized standard score differences clearly illustrated the inverse distribution between scales of total FIS-40, total COMPASS-31, and total PSQI that attribute higher scores to more severe symptoms and SF-36 subscales that do the opposite. As a result, our cohort of 250 cases was subdivided into cluster 1, including cases showing more severe symptoms in all five selected parameters (*n* = 94), cluster 2 with cases presenting moderate affection (*n* = 107), and a smaller group of only 49 individuals with milder symptoms (cluster 3) (Figure 1). This shows that the cohort studied mostly contained severe to moderate cases, with <20% of mildly affected cases. The definition of severe in cluster 1 involve total FIS scores over 145 on average, scores over 65 for total COMPASS-31, scores over 15 on average for PSQI, and the lowest scores for physical functioning and bodily pain, which may translate into a more severe fatigue phenotype, accompanied by dysautonomia and sleep problems while experiencing higher levels of pain and compromised physical functioning than the other two clusters (Table 2, Figure 1).

### 3.3. Cluster-Based Differential Analysis of Blood Parameters 

Next, we assessed potential differences in the blood analytical variables of these three clusters of cases (clusters 1, 2 and 3, as defined above) using univariate analyses (see Section 2 for details). The analyses intended to detect blood parameters within normal reference range values presented significant differences across case groups (clusters) and may, therefore, be associated with case symptoms. The statistical analysis detected five blood parameters fulfilling the requirements, with *p*-values < 0.05 (Table 3).

However, when looking at statistical differences between individual pair sets, we found that none of these five parameters could individually differentiate clusters 1, 2, and 3. For hemoglobin (Hb) levels, only the moderate group (cluster 2) differentiated from the other two, an observation with unclear physiologic interpretation. For neutrophil counts (NT), differences were found between severe (cluster 1) and mild (cluster 3) cases, but none were detected for the moderate group. For cholesterol levels (COL), severe cases (cluster 1) presented differences with moderate and mild (clusters 2 and 3), with no differences between the latter two, whereas high-density lipoproteins (HDL) differences appeared between clusters 1 and 2, but not with cluster 3. Lastly, the levels of complement factor 3 (C3) presented differences between the severe and mild clusters, as well as between the mild and the moderate, but no differences were detected between the severe and the moderate clusters, indicating potential value as a marker to differentiate mild cases from the rest (Figure 2).

In conclusion, symptom-based case clustering followed by differential blood analytics was inefficient for detecting robust single blood variables correlated with case health severity, as defined in these three clusters. It is, however, interesting that some of these blood parameters could, to some extent, differentiate between clusters, the significance of which is not understood at present.

### 3.4. Stratified Analysis

As an additional attempt to detect patient subgroups that could help refine current diagnosis methods, we evaluated sets of cases presenting abnormal blood parameter values for stratification purposes.

#### 3.4.1. Outstanding Blood Parameters with Abnormal Values

Top analytic variable values deviating from established healthy population reference ranges were vit D (60.4%) mostly represented by deficiency, LDL (55.6%), complement factor C1q (42.8%), and cholesterol (26.4%), all showing increased values overall. Platelet mean volume values appeared both increased and decreased (Table 4 and Appendix A).

Combinations of these five blood analytical variables which included at least 20% of the cases in our cohort showed that vitamin D and LDL were both abnormal in 32.4% of the participants. Other combinations, such as vitamin D deficiency and increment of C1q, or increased LDL with C1q or cholesterol involved over 20% of the participants (>50 cases). Since vitamin D reference values are widely influenced by genetic and environmental factors, and the assays to quantitate its levels typically show variability over ±10% [20,21], leading to a lack of consensus global reference values [22,23], we decided to exclude this variable from our stratified downstream analysis.

Similarly, since differences found in LDL and cholesterol are nonspecifically associated with disease, sometimes appearing together with vitamin D deficiency [22,23], they were not further pursued, meaning that they were not used to set stratification conditions of our cohort. The unexpected finding of a quite significant proportion of ME/CFS cases showing increased levels of C1q and decreased C1 inhibitor (42.8% and 8.8%, respectively) (Appendix A), together with the C1q deficiency being associated with autoimmune diseases such as systemic lupus erythematosus [24], motivated our interest to hypothesize C1q as a potential biomarker for ME/CFS subtyping.

#### 3.4.2. Symptom Differences across C1q Case Clusters

Downstream analysis after a conservative 5% cutoff above C1q maximum normal value stratification (samples with C1q > 26.05 mg/dL, cluster 1, with *n* = 90; samples with C1q < 26.05 mg/dL, cluster 2, with *n* = 160), however, showed no significant differences in any symptom score, with the only exception of a tendency for bodily pain (*p* = 0.09), suggesting perhaps increased pain in the group with high C1q levels (mean values were 14.24 vs. 17.33, respectively) (Table 5).

#### 3.4.3. Blood analytic Differences across C1q Case Clusters

To evaluate whether the group of cases presenting increased C1q levels (cluster 1, *n* = 90) could, at the same time, present with additional blood parameter differences, even when being within normal reference values for the group showing normal C1q values (cluster 2, *n* = 160), we applied a statistic test after cohort stratification, as detailed in Section 2. The results showed significant differences in seven blood parameters (*p* < 0.05), as shown in Table 6.

Overall, the cluster with increased C1q levels presented with higher red blood cell counts, as well as total protein, C3, and C4 levels, and lower IgG3, IgG4, and C1_inh_ concentrations, indicating pathways potentially connected with elevated C1q levels and, thus, potentially relevant for clinical treatment of an important subset of ME/CFS cases.

## 4. Discussion

As earlier mentioned, the use of standard blood tests to at least support a potential case of ME/CFS (“triage” diagnosis method) and/or differentiate case subgroups for therapeutic and research purposes would provide clear advantages. In fact, it may well constitute the key toward unveiling ME/CFS subgroup etiology and evolution.

Although clustering methods based on case symptoms have been useful in identifying autonomic phenotypes in CFS [25], they failed at detecting robust blood correlations between symptoms and individual blood parameters in our cohort (Figure 2). The approach did, however, show potential for differentiating cases with severe, moderate, or mild affection as defined by the five symptom scores used for clustering (Table 2 and Table 3). The physiological significance of the findings, including the increased levels of C3 in severe and moderate with respect to mildly affected cases or the increased neutrophil count in severe ME/CFS by itself or in combination with LDL and cholesterol, as well as their involvement in symptom development or maintenance, remains to be elucidated.

Nevertheless, the presence of a large proportion of ME/CFS (42.8% or 107/250) cases with increased expression of C1q, for the first time, may importantly set the basis for future ME/CFS subtyping. 

C1q acts as the first component in the classical complement pathway. The complement system is a central part of innate immunity with two important functions: serving as a defense system against invading pathogens and for the clearing of dead cells or debris [26,27]. C1q recognizes PAMPs (pathogen-associated molecular patterns), including LPS (lipopolysaccharide) and bacterial porins [28], in addition to recognizing molecules such as phosphatidylserine and dsDNA exposed on the surface of dying cells [29,30].

Thus, the detected increased levels of C1q and other complement components in a subgroup of ME/CFS cases may indicate a state of active efferocytosis toward fighting a subjacent infection or while clearing damaged tissue. Moreover, cases with chronic activation of the complement pathway may, for this reason, become particularly sensitive to PEM [4,5,6,7], a possibility that may be worth exploring. 

It is well documented that, both inefficient and overstimulation of the complement system can be detrimental for the host, being associated with increased susceptibility to infections, autoimmunity, chronic inflammation, and thrombotic microangiopathy, among others [24,26,27,31]. Some of these processes have been associated with ME/CFS [32,33]. We, however, only found a few cases of positive self-antigen immunity across the 10-test run applied to the 250 participating cases (Appendix A).

The observation that, among the many blood parameters measured, those known to be related to C1q function, i.e., C1 inhibitors C3 and C4 showed significant differences between groups (Table 6) further supports a functional problem of the complement system in this subgroup of cases, perhaps with consequences in the process of coagulation. A prospective follow-up of coagulopathies in this subgroup of patients, thus, appears pertinent.

More recently, Benavente et al. showed that C1q acts as a ligand that can directly bind a series of receptors previously unidentified as partners of this molecule, including the following proteins: CD44, GPR62, BAI1, c-MET, and ADCY5, which trigger activation of downstream signaling pathways [34,35] and, thus, affect different aspects of neuroepithelial stem-cell biology. The finding by these authors that C1q is elevated upon nerve injury [34] and the emerging connections of C1q with neurodegenerative disease [36,37] open up the exciting possibility that increased levels of C1q may underlie ME/CFS cognitive problems [4,5,6,7]. C1q alters prion disease progression, regulates neuron pruning, and modulates the process of phagocytosis by microglia while responding to amyloid plaque formation [38,39,40]. Unfortunately, no specific instruments for detailed cognitive assessment of the participants or neuroimaging tools were used in this study.

Lastly, the fact that the two strata presenting normal vs. increased levels of C1q showed differences in pain may indicate a direct involvement of C1q in case symptoms. It may be relevant to include a more detailed assessment of this symptom by using pain-focused questionnaires, such as the FIQ (Fibromyalgia Impact Questionnaire) and/or others [41,42,43], in future studies of C1q’s role in ME/CFS. Although the failure of questionnaires other than the SF-36 to detect symptom-related differences with C1q levels (Table 5) lead to us presuming no major involvement of C1q in this aspect of the disease, it seems curious that the plotting of the two clusters showed opposite trends in all five symptoms selected by the k-means screening method to set symptom-based clusters of the cohort (Figure 1 and Figure 3).

Moreover, within the 94 cases in the “severe” group (cluster 1) of our symptom-based cluster analysis (Table 2), about 39% (37/94) presented increased C1q levels while 61% (57/94) showed normal levels; within the 107 cases of the “moderate” group (cluster 2), 35% (37/107) had increased C1q and 65% (70/107) normal C1q levels; within the 49 cases of the “mild” group (cluster 3), 33% (16/49) showed increased C1q levels and 67% (33/49) had C1q levels within normal reference values. This indicates a rough 1:2 overall ratio of cases with increased C1q levels in the “mild” group, with a slight increase in this ratio in the “moderate” and an even higher ratio (1:1.5) in the “severe” cluster, suggesting an increased prevalence of high C1q levels with case disease severity, despite the lack of significant correlations between individual symptom scores and C1q levels. A correlation of C1q levels with the chronicity status of cases could not be established either.

Additional blood parameters that showed abnormal values in a large proportion of individuals in our cohort, such as vitamin D deficiency, LDL levels, cholesterol, C3 levels, or platelet mean value, may provide relevant information for treatment options, an aspect not well understood at present, which requires further monitoring in the future.

It should be mentioned that, although different reference values have been established for vitamin D deficiency, for example, the recommendations from the 2011 US Institute of Medicine (IOM) reported a minimal concentration of 52 nmol/L, while the US Endocrine Society guidelines stated a minimal concentration of 78 nmol/L [44,45], the applied range in this study was right below the lowest range (50 nmol/L, Appendix A), and yet a large proportion of cases (>60%) showed vitamin D deficiency (Table 4).

### Additional Limitations

Although the cohort under study included a considerable number of subjects (*n* = 250), the external validity of the data remains limited to females. Random selection of participants can lead to more representative results population wise; however, subject heterogeneity translates into enhanced variability, compromising the establishment of robust differences, perhaps relevant to the diagnosis of ME/CFS. The lack of additional relevant differences across ME/CFS cases cannot be ruled out by the laborious, yet discrete analysis here performed.

## 5. Conclusions

In conclusion, this study identified a potential new player in the ME/CFS pathology, the C1q component of the complement system, affecting over 40% of cases. This finding paves the way for exploring a C1q-based standard lab assay to detect ME/CFS subtypes with relevant clinical and research implications. The understanding of the underlying pathomechanisms behind this finding is limited at present, granting further exploration of the observation.

## Figures and Tables

**Figure 1 jcm-10-04171-f001:**
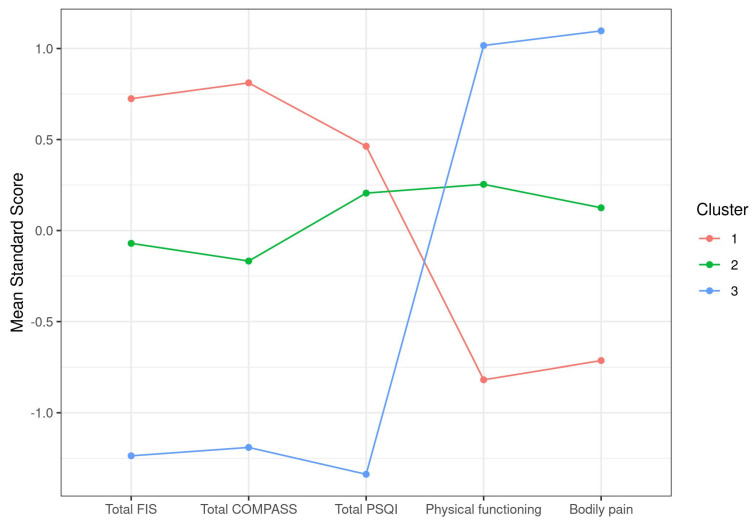
Graphic representation of ME/CFS clustering according to symptom standard mean scores.

**Figure 2 jcm-10-04171-f002:**
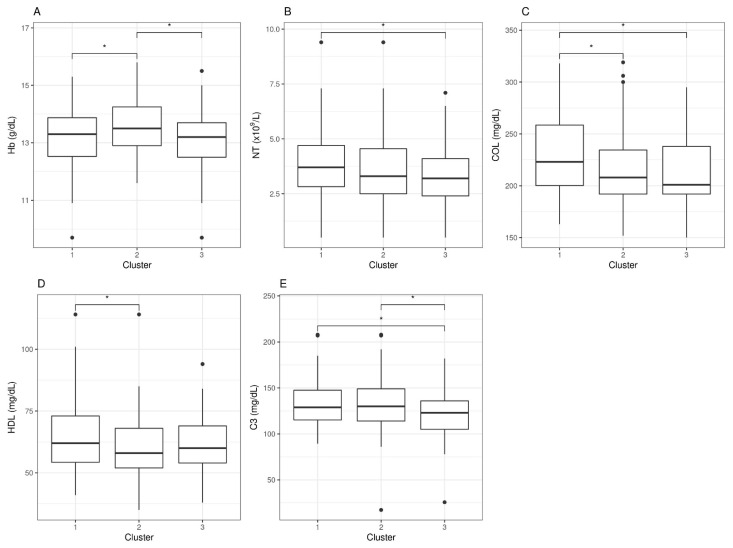
Blood analytic difference boxplots between ME/CFS symptom-based clusters. Abbreviations: Hb, hemoglobin; NT, neutrophil counts; COL, cholesterol; HDL, high-density lipoprotein; C3, complement 3. The significance level was set at * *p* < 0.05. Data beyond 1.5 inter-quartile range values, representing potential outliers, are plotted as individual dots.

**Figure 3 jcm-10-04171-f003:**
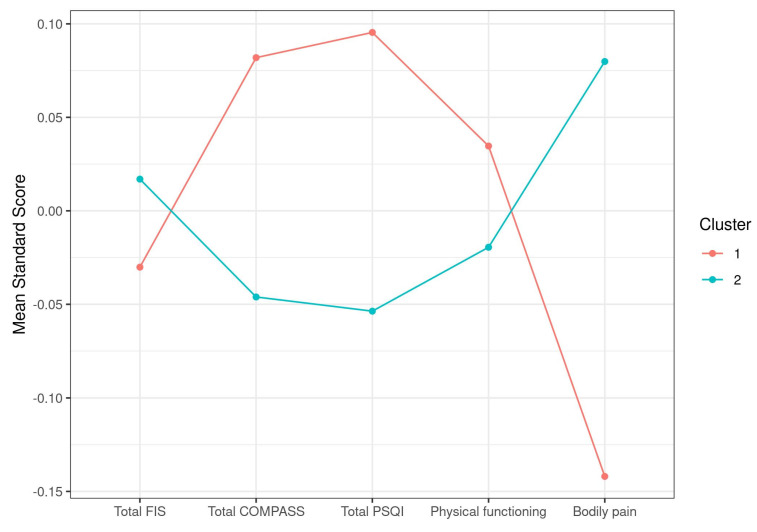
Graphic representation of ME/CFS symptom standard score differences in relation to C1q stratification.

**Table 1 jcm-10-04171-t001:** Demographics and clinical characteristics of participants at baseline. Data are expressed as the mean ± standard deviation (SD) for continuous variables and compared by Student *t*-test, whereas categorical variables are given as numbers with percentages (%) and compared by Fisher’s exact test.

Variables	ME/CFS (*n* = 250)
Age, years	45.9 ± 7.02
BMI, kg/m^2 †^	24.5 ± 4.72
SBP, mmHg	125.8 ± 2.5
DBP (mmHg)	76.3 ± 1.6
**Medication, *n* (%)**	
NSAIDs	9 (42.9)
Hypnotics	5 (23.8)
Antidepressants	6 (28.6)
Antipsychotics	4 (19.0)
Opioids	11 (52.4)
**Measures**	
**FIS-40**	
Global score (0–160)	
Physical	35.4 ± 2.4
Cognitive	34.0 ± 3.4
Psychosocial	63.9 ± 2.4
**COMPASS-31**	
Global score (0–100)	53.6 ± 3.5
Orthostatic intolerance	24.3 ± 2.1
Vasomotor	1.4 ± 2.7
Secretomotor	9.3 ± 3.4
Gastrointestinal	11.6 ± 2.9
Bladder	3.5 ± 4.1
Pupillomotor	3.7 ± 3.4
**PSQI**	
Global score (0–21)	14.0 ± 0.7
Subjective sleep quality	1.9 ± 0.1
Sleep latency	2.2 ± 0.1
Sleep duration	1.5 ± 0.1
Habitual sleep efficiency	1.9 ± 0.2
Sleep disturbances	2.4 ± 0.1
Sleeping medication	1.9 ± 0.2
Daytime dysfunction	2.2 ± 0.1
**SF-36**	
Physical functioning	26.9 ± 0.6
Physical role	3.7 ± 0.81
Bodily pain	16.2 ± 1.55
General health perception	21.3 ± 2.18
Vitality	17.0 ± 1.58
Social role functioning	28.2 ± 1.87
Emotional role functioning	30.5 ± 2.78
Mental health	41.4 ± 3.12

Abbreviations: BMI, body mass index; DPB, diastolic blood pressure; SBP, systolic blood pressure; FIS-40, 40-item Fatigue Impact Scale; COMPASS-31, 31-item Composite Autonomic Symptom Score; PSQI, Pittsburgh Sleep Quality Index; SF-36, 36-item Short-Form Health Survey; NSAIDs, nonsteroidal anti-inflammatory drugs. ^†^ The body-mass index (BMI) is the weight in kilograms divided by the square of the height in meters.

**Table 2 jcm-10-04171-t002:** Clustering of ME/CFS cases according to symptom differences, as supported by k-means analysis. Data are presented as the mean ± SD for each item. Physical functioning and bodily pain were evaluated by two items of the 36-item Short-Form Health Survey (SF-36).

	Cluster 1	Cluster 2	Cluster 3	*p*-Value
Total FIS	147.93 (9.88)	131.79 (13.48)	108.1 (21.83)	<0.0001
Total COMPASS	66.82 (10.52)	50.9 (10.77)	34.26 (12.71)	<0.0001
Total PSQI	15.82 (3.4)	14.75 (2.94)	8.33 (2.78)	<0.0001
Physical functioning	12.82 (8.82)	31.27 (12.75)	44.39 (16.79)	<0.0001
Bodily pain	6.3 (7.77)	17.95 (11.79)	31.45 (11.83)	<0.0001
Size	94	107	49	

Standard deviation values are shown in parentheses.

**Table 3 jcm-10-04171-t003:** Blood analytic differences between symptom-based case clusters. Data are presented as mean ± (SD) for each biochemical variable.

	Cluster 1	Cluster 2	Cluster 3	*p*-Value
Hb (g/dL)	13.16 (1.05)	13.57 (0.94)	13.07 (1.12)	0.0033
NT (×$10^{9}$/L)	3.99 (1.65)	3.58 (1.63)	3.3 (1.48)	0.0365
COL (mg/dL)	229.57 (36.69)	216.6 (37.42)	211.84 (33.73)	0.0077
HDL (mg/dL)	64.71 (14.44)	59.71 (12.48)	63.12 (12.65)	0.0292
C3 (mg/dL)	132.32 (24.65)	132.02 (29.96)	119.89 (28.09)	0.0246

Abbreviations: Hb, hemoglobin; NT, neutrophil counts; COL, cholesterol; HDL, high-density lipoprotein; C3 complement factor 3.

**Table 4 jcm-10-04171-t004:** Top blood analytic variables showing abnormal values with respect to reference values in our cohort (*n* = 250). The number of cases with abnormal values with respect to reference values and percentages (%) are shown.

Variables	*n* (%)
25(OH).Vit.D3	151 (60.4)
LDL	140 (56)
C1q	107 (42.8)
25(OH).Vit.D3, LDL	81 (32.4)
25(OH).Vit.D3, C1q	72 (28.8)
COL	66 (26.4)
C1q, LDL	60 (24)
COL, LDL	58 (23.2)
PMV	56 (22.4)

Abbreviations: 25(OH) Vit.D3, vitamin D; LDL, low-density lipoprotein; C1q, complement factor C1q; COL, cholesterol; PMV, platelet mean volume.

**Table 5 jcm-10-04171-t005:** Symptom differences between case clusters with increased C1q values (cluster1) (>26.05 mg/dL) or within range (C1q *<* 26.05 mg/dL). Group mean values and standard deviations are shown.

	Cluster 1	Cluster 2	*p*-Value
Total FIS	132.6 (22.83)	133.56 (18.81)	0.7358
Total COMPASS	54.96 (16.84)	52.88 (15.94)	0.3401
Total PSQI	14.29 (4.06)	13.67 (4.21)	0.2544
Physical functioning	27.5 (18.36)	26.57 (16.56)	0.6907
Bodily pain	14.24 (13.63)	17.32 (13.95)	0.0906
Size	90	160	

**Table 6 jcm-10-04171-t006:** Blood analytic differences between C1q case clusters. Group means and standard deviations are shown.

	Cluster 1	Cluster 2	*p*-Value
RBC (×$10^{12}$/L)	4.62 (0.32)	4.53 (0.38)	0.0431
PT (g/dL)	7.24 (0.39)	7.1 (0.42)	0.0093
IgG3/IgG	6.29 (3.64)	8.91 (13.47)	0.0219
IgG4/IgG	2.75 (1.7)	3.32 (2.56)	0.0343
C1*_inh_* (mg/dL)	25.56 (5.28)	27.56 (5.58)	0.0055
C3 (mg/dL)	137.44 (27.47)	125.43 (27.48)	0.0011
C4 (mg/dL)	30.73 (8.57)	27.7 (7.72)	0.006

Abbreviations: RBC, red blood cell; PT, total protein; IgG, immunoglobulin; C1_inh_ complement 1 inhibitor; C, complement factor. Standard deviation values are shown between brackets.

## Data Availability

Datasets are provided as Appendix A.

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
