# Peer review of "Complement Component C1q as a Potential Diagnostic Tool for Myalgic Encephalomyelitis/Chronic Fatigue Syndrome Subtyping"

_jcm, 2021, doi:10.3390/jcm10184171_

Round 1

Reviewer 1 Report

Well done!

Author Response

Thank you

Reviewer 2 Report

The paper has obviously been previously reviewed due to the corrects. The crux issue being reported in the mean complement ICq levels. What one really needs in using the data at a clinical level is a value that can be assessed in patients. To achieve that you need to present the frequency of subjects above or below a definable level. The graph implies that only a small number of cases are likely to be abnormal compared with the other groups. Whilst the mean/median may be different the we need the outliers identified.

Author Response

Thank you for the time taken to review the manuscript. In reference to abnormal C1q levels, we used the normal reference values of 10-25 mg/dL as provided by the Vall d’Hebron University Hospital, Barcelona, Spain Lab facility. Since normal reference levels may depend on the test applied, we have added in the Methods section of the Manuscript the method used by the Hospital´s lab (nephelometry) and the instrument (BN II System, Siemens Healthcare Headquarters Erlangen, Germany) to determine complement levels in patients´ serum (page 3, lines 137 and following). Because this is the first report of a high frequency of ME/CFS cases presenting increased C1q levels, we feel it is premature to establish or recommend a classification or subtyping of ME/CFS patients according to particular C1q levels. This is the reason we do not give a defined value to use in the clinic, instead we just propose that this variable may result helpful in the future and recommend measuring it in additional cohorts “The finding paves the way for exploring a C1q-based standard lab assay to detect ME/CFS subtypes with relevant clinical and research implications” (page 12, lines 402 and following).  With respect to a graph showing  “only a small number of cases are likely to be abnormal value” referred by the reviewer, we respectfully disagree that the number was small. As shown (plotted graph) on page 3 of Supplementary Figure S1, and as listed on Table 4, 42.8 % of cases (107 out of 250) showed increased C1q values, taken as increased values those above the maximum normal level (25 mg/dL). If the cutoff normal value is increased by a 5% (26.5 mg/dL) still 90 of the 250 patients (36%) would show increased levels (explained on page 9, lines 280 and following). We apologize for not having provided a Figure legend for Supplementary Figure S1 which has now been added, as follows: “ Supplementary Figure S1. Box plots of top blood analytic variables showing off-normal reference values in our cohort (n = 250). Red lines indicate normal range values. Black line within the box is the median within quartile values. Normal values are shown in green while abnormal values are shown in red. “

This manuscript is a resubmission of an earlier submission. The following is a list of the peer review reports and author responses from that submission.

Round 1

Reviewer 1 Report

The manuscript by Castro-Marrero, Complement component C1q as a potential diagnostic tool for Myalgic Encephalomyelitis/Chronic Fatigue Syndrome subtyping is generally well-written and scientifically sound. This work provides useful information to clinicians in the field of ME/CFS. I have only minor suggestions that should be addressed to make the manuscript better.

  1. The manuscript would benefit from a primary English speaker to proof it. There are a few instances of incorrect grammar that make reading awkward but this is only a minor issue.
  2. There are some minor technical writing issues. The most salient one is the use of counting numbers. Numbers less then 9 should be spelled out unless they contain unit modifiers or are in a series. For instance, from, line 125 "within 2 hours should be within two hours" if it was 2-hr then it would be fine. However, from line 102, orthostatic intolerance (4 items), vasomotor (3 items)" this is correct. I suggest you check with the ACS style  guide if you are unsure. In some places its correct but not in others.
  3. Finally, it is more appropriate to refer to your subjects as 'cases" in the context of "cases and "controls", not "patients". If it were a case study of the subjects were actual patients being treated then the term "patient" would be appropriate. This is a common mistake.

Author Response

Reviewer #1

The manuscript by Castro-Marrero, Complement component C1q as a potential diagnostic tool for ME/CFS subtyping is generally well-written and scientifically sound. This work provides useful information to clinicians in the field of ME/CFS. I have only minor suggestions that should be addressed to make the manuscript better.

  1. The manuscript would benefit from a primary English speaker to proof it. There are a few instances of incorrect grammar that make reading awkward but this is only a minor issue.

Response: Thank you very much for your appraisement and for the compliments. To improve the grammar, we have reviewed some sentences throughout the manuscript without altering their meaning.

  1. There are some minor technical writing issues. The most salient one is the use of counting numbers. Numbers less than 9 should be spelled out unless they contain unit modifiers or are in a series. For instance, from line 125 "within 2 hours” should be “within two hours" if it was 2-hrs then it would be fine. However, from line 102, orthostatic intolerance (4 items), vasomotor (3 items) this is correct. I suggest you check with the ACS style guide if you are unsure. In some places its correct but not in others.

Response: Thank you for the information. We have now applied the indicated rule to comply with the ACS style guide on the pointed examples and have reviewed the complete manuscript to apply the rule where needed.

  1. Finally, it is more appropriate to refer to your subjects as “cases" in the context of "cases and controls", not "patients". If it were a case study of the subjects were actual patients being treated, then the term "patient" would be appropriate. This is a common mistake.

Response: Thank you for the information. The term “patient(s)” has been replaced by “case(s)” where appropriate in the whole manuscript.

Reviewer 2 Report

I think that the research is original and interesting

Author Response

Reviewer #2

I think that the research is original and interesting.

Response: Thank you very much for your appraisement and for the compliments.